# What Are the Biomechanical Properties of an Aortic Aneurysm Associated with Quadricuspid Aortic Valve?

**DOI:** 10.3390/jcm11164897

**Published:** 2022-08-20

**Authors:** Siyu Lin, Marie-Catherine Morgant, Diana M. Marín-Castrillón, Chloé Bernard, Arnaud Boucher, Benoît Presles, Alain Lalande, Olivier Bouchot

**Affiliations:** 1IFTIM, ImViA Laboratory, University of Burgundy, 21078 Dijon, France; 2Department of Cardio-Vascular and Thoracic Surgery, University Hospital of Dijon, 21078 Dijon, France; 3Department of Medical Imaging, University Hospital of Dijon, 21078 Dijon, France

**Keywords:** ascending aortic aneurysms, quadricuspid aortic valve, biomechanical properties, histology

## Abstract

Association of quadricuspid aortic valve (QAV) with ascending aortic aneurysms (AsAA) is rare. A 63-year-old female with hypertension was found (on MRI) to have an ascending aortic aneurysm (52 mm in maximum diameter) and dilatation at the level of the sinotubular junction (38 mm in diameter) associated with quadricuspid aortic valve. An ascending aortic wall replacement surgery was performed. In this study, we focus on the behavior of the aorta associated with QAV considering the in vitro biomechanical characteristics and histology. The properties of QAV are closer to bicuspid aortic valve than tricuspid aortic valve, but with higher wall thickness.

## 1. Introduction

Ascending aortic aneurysm (AsAA) is a life-threatening pathology causing a permanent dilatation associated with a high risk of aortic dissection that may result in the death of the patient. According to the European guidelines, AsAA surgery consists of replacement of the aorta with synthetic grafts [1]. The decision to operate is made based on the maximum diameter of the aorta (larger than 55 mm for patients with no presence of bicuspid aortic valve or Marfan syndrome).

A normal aortic valve has three leaflets. The aortic valve has two leaflets in 1.3% of the population, known as bicuspid valve (BAV) [2]. In less than 0.0004% of the population, the aortic valve has four leaflets, called quadricuspid aortic valve (QAV) [3]. An ascending aneurysm is associated with bicuspid valve in 30% of cases, while it is rare to observe an AsAA associated with quadricuspid valve. The biomechanical properties of AsAA stay complex due to the anisotropic character and heterogeneity along the main axis of the artery [4]. This article introduces a study concerning the biomechanical behavior of AsAA with quadricuspid aortic valve (AsAA-QAV) based on preoperative magnetic resonance imaging (MRI), in vitro mechanical study of the aortic wall, and histology.

## 2. Population

This study, recruiting patients with AsAA replacement for biomechanical tests, was approved by the national ethics committee (2018-A02010-55). The study was recorded on ClinicalTrials.gov (clinical registration number NCT03817008). No written consent was required by French law. Each patient received written information about the study in accordance with the ethics committee. Upon receiving oral consent, the patient was provided with written information explaining the study’s principles and course. An attestation of the patient’s oral consent was signed by the investigating physician and countersigned by the patient.

Among 100 patients in our database, the choice of the comparison was made earlier on the aortic diameter. A 63-three-year-old female with hypertension had an aneurysm at the level of the thoracic ascending aorta (maximum diameter of 52 mm on MRI) and dilatation at the level of the sinotubular junction (38 mm for the maximum diameter of AsAA) associated with QAV. To compare, one patient with BAV (53 mm maximum diameter of AsAA) and one patient with tricuspid aortic valve (53 mm maximum diameter of AsAA) were chosen based on similar clinical characteristics (Table 1). 

## 3. Method 

Samples of the aortic wall were obtained from aortic replacement surgeries. The in vitro experimental study started within 30 min after the aortic replacement. The aortic wall samples were preserved in phosphate-buffered saline during the transfer from the operating room to the laboratory for biaxial tensile experiment. The aortic walls were cut into squares (15 mm × 15 mm). According to the localization, the AsAAs were divided into medial (MED), posterior (POST), lateral (LAT), and anterior (ANT) quadrants. According to the blood direction, each specimen was marked circumferential (CIR) and longitudinal (LON) (Figure 1). The biomechanical experiments were carried out using a biaxial tensile test machine (LM1 Planar Biaxial, TA Instruments, New Castle, DE, USA). Each specimen was placed in 10× 10% preconditioning to deform the contraction caused by the sectioned aorta. A further stretch at a rate of 10 mm/min was programmed till rupture. Five points were marked on each specimen to track the in-plane movement with a digital camera (Prosilica GE, Allied Vision Technologies, Stadtroda, Germany). Average thickness was measured with the help of an electronic micrometer (Litematic VL-50, Mitutoyo, Sakado, Japan). 

Stresses and strains were determined by force and length. Engineering strain indicated the ratio of deformed length _∆_*l* divided by resting specimen length (after preconditioning) *l*_0_, where _∆_*l* is the difference between the resting specimen length and the load-filled specimen length *l*. Stress *σ* refers to the amount of tensile load *F* recorded during the test per unit loaded cross-sectional area *A* of the specimen. Aortic tissue was defined as incompressible. Area *A* was equal to the resting-specimen cross-sectional area *A*_0_, where *A*_0_ can be computed from the load-free specimen thickness and length *l*_0_.

Based on the information of real-time load and displacement obtained by the biaxial tensile, as well as the calculated average thickness, the maximum value of Young’s modulus for the CIR and LON directions corresponded to the ratio between the stress and the strain and described the failure stiffness before the rupture of the aortic wall.

The histological samples were resected after aortic wall collection before the biaxial tensile test. To perform a histological analysis, the AsAA segments were fixed in 10% neutral formalin buffer for 48–72 h and stored in 70% ethanol. To quantify collagen and elastin, each piece of tissue was cut into 5 μm sections and stained with Masson trichrome and Verhoef stain.

## 4. Discussion

The most striking finding was that based on failure stiffness and histology, the biomechanical properties of MED and LAT quadrants on AsAA-QAV were similar to AsAA-BAV compared with AsAA-TAV. In terms of failure stiffness, the elastic modulus showed similar values for the three patients: the lateral quadrant of the aorta showed a higher stiffness (1.062 ± 0.138 MPa), roughly double the value of the medial quadrant (0.687 ± 0.103 MPa). The CIR direction of the aorta was stiffer than the LON direction, except for the LAT quadrant of both AsAA-BAV and AsAA-QAV (Table 2). 

To be more specific, in the MED quadrant, the failure stiffness of CIR was greater than that of the LON direction. Moreover, the failure stiffness in the CIR direction of AsAA-TAV was higher than that of other AsAA samples in other directions. In the LAT quadrant, the CIR showed a higher value than in the LON direction only in the AsAA-TAV sample. However, in the LAT quadrant of both AsAA-BAV and AsAA-QAV samples, the LON direction had significantly greater failure stiffness than the CIR direction. In a previous study [5] on 100 patients, the value of failure stiffness in the LAT quadrant was the highest among all regions, while the value of the MED quadrant was the lowest. Therefore, the stress–strain and Young’s modulus–stress curves of MED and LAT quadrants were chosen for comparison (Figure 2). BAV and QAV showed similar behaviors in both CIR and LON directions for the MED and LAT quadrants. From the thickness point of view, the AsAA-QAV was thicker than the two other two cases (Table 2).

In terms of histology, AsAA-QAV indicated less collagen (Figure 3). Indeed, the content of collagen was higher in AsAA-TAV than in AsAA-BAV and AsAA-QAV. Moreover, the content of elastin of AsAA- QAV showed a similarity to AsAA-BAV and a higher value than for AsAA-TAV (Table 3). 

With the uniaxial tensile test, it was found that AsAA in the CIR direction was stiffer than in the LON direction [6]. It was also found that there was no significant change in the thickness of the aortic wall between AsAA-TAV and AsAA-BAV. In a previous study using biaxial testing, we found the same results in AsAA associated with BAV and TAV [7]. In this study, AsAA-QAV matched the range of thickness and biomechanical properties. This might indicate that the valve type did not change the values of stiffness, which might be related to the similar orientation of the fiber. Meanwhile, it was shown that the medial quadrant had less stiffness whatever the type of valve. 

According to the study of Bersi et al. [8], local stiffness has a positive relationship with elastin and a negative relationship with collagen. The aorta with TAV has a higher level of collagen and a lower level of elastin [9]. According to the study by Hosoda et al. [10], the range of collagen and elastin in the human aorta is 19% to 30% and 17% to 40%, respectively. The amount of elastin decreased in the AsAA [11]. In our study, the collagen content in the AsAA-BAV and AsAA-QAV had a lower percentage than AsAA-TAV, while the elastin content in the AsAA-TAV was higher than the other two types of the aorta.

This study has two major limitations. The first limitation is the small sample: only three patients were selected for comparison. Indeed, quadricuspid valve is a very rare anomaly, preventing having a significant patient database. Furthermore, due to the recommendation of the current surgical guidelines for AsAA replacement, the main inclusion criterion was the aortic diameter rather than other risk factors, such as hypertension, obesity, and aortic valve disorder. These factors were not considered for the selection of the patients.

The most striking finding is that based on failure stiffness and histology, the biomechanical properties of MED and LAT quadrants on AsAA-QAV are similar to AsAA-BAV compared with AsAA-TAV. Meanwhile, the thickness of AsAA-QAV is higher than both AsAA-BAV and AsAA-TAV.

## 5. Conclusions

In this study, the aortic biomechanical properties of quadricuspid aortic valve were similar to those of bicuspid aortic valve, but relatively different from those of tricuspid aortic valve. Moreover, the thickness of the ascending aortic aneurysms of quadricuspid aortic valve showed a higher value than the two other types of valves. 

## Figures and Tables

**Figure 1 jcm-11-04897-f001:**
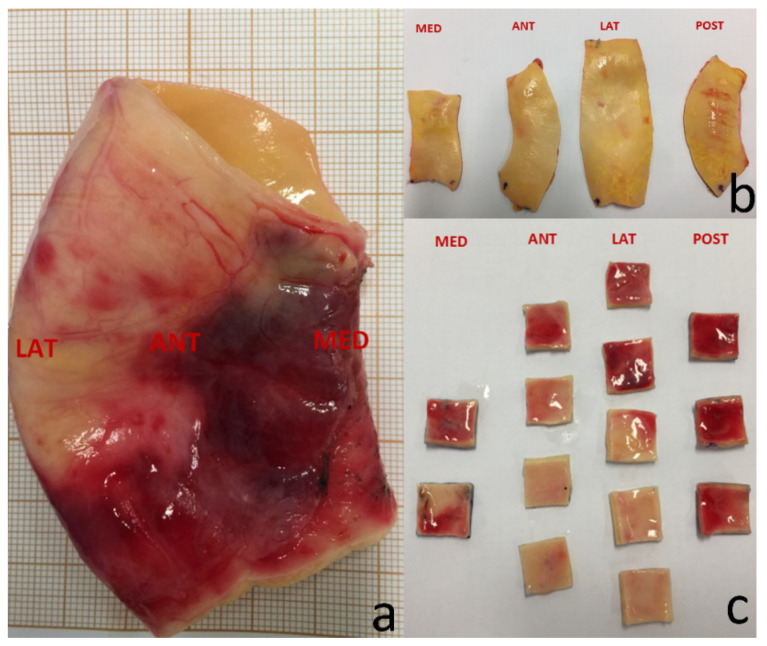
Preparation of the aortic samples for in vitro biomechanical experiments. (**a**) Global view of the ascending aorta. (**b**) Quadrant view from the aortic internal according to MED, ANT, LAT, POST. (**c**) Specimen view from the aortic external. MED = medial; POST = posterior; LAT = lateral; ANT = anterior.

**Figure 2 jcm-11-04897-f002:**
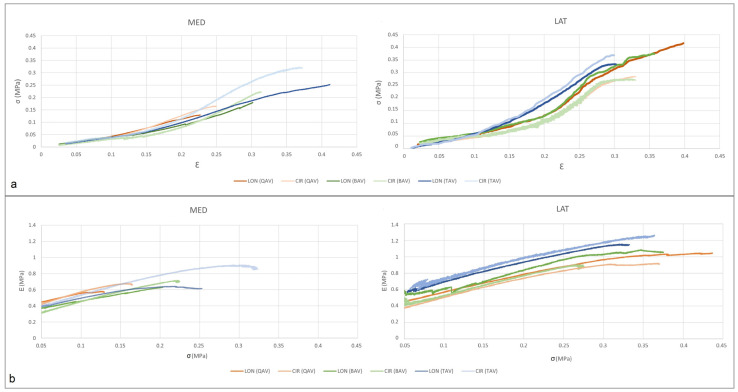
Stress–strain and Young’s modulus–stress curves on the AsAA associated with QAV, BAV, and TAV. (**a**) stress–strain curves of MED and LAT quadrants; (**b**) Young’s modulus–stress curves of MED and LAT quadrants. *σ *= stress; *Ԑ* = strain; *E =* Young’s modulus; MED = medial; LAT = lateral; QAV quadricuspid aortic valve; BAV = bicuspid aortic valve; TAV = tricuspid aortic valve.

**Figure 3 jcm-11-04897-f003:**
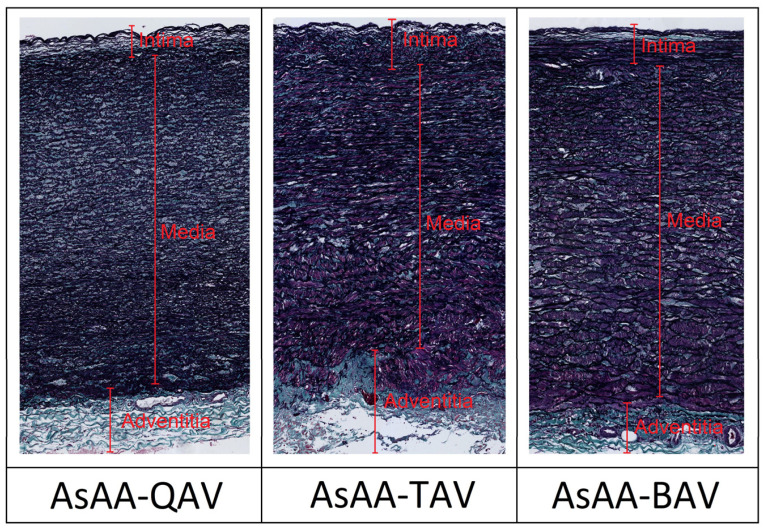
Histological images with Masson trichrome and Verhoef staining for the aortic wall with each type of valve. AsAA-QAV = ascending aortic aneurysms associated with quadricuspid aortic valve; AsAA-TAV = ascending aortic aneurysms associated with tricuspid aortic valve; AsAA-BAV = ascending aortic aneurysms associated with bicuspid aortic valve.

**Table 1 jcm-11-04897-t001:** Clinical characteristics of the three patients with ascending aortic aneurysm replacement. AsAA-QAV = ascending aortic aneurysms associated with quadricuspid aortic valve; AsAA-TAV = ascending aortic aneurysms associated with tricuspid aortic valve; AsAA-BAV = ascending aortic aneurysms associated with bicuspid aortic valve.

	AsAA-QAV	AsAA-TAV	AsAA-BAV
**Aortic diameter (mm)**	52	53	53
**Age (years)**	63	44	58
**Gender**	Female	Male	Male
**Tobacco use**	−	−	−
**Dyslipidemia**	−	−	−
**Hypertension**	+	+	−
**Obesity**	−	+	+
**Diabetes**	−	−	−
**Coronary arterial disease**	−	−	−
**Aortic insufficiency**	+	−	−
**Aortic stenosis**	−	−	+

**Table 2 jcm-11-04897-t002:** The failure stiffness (FS) in circumferential and longitudinal directions and the thickness in different quadrants. AsAA-QAV = ascending aortic aneurysms associated with quadricuspid aortic valve; AsAA-TAV = ascending aortic aneurysms associated with tricuspid aortic valve; AsAA-BAV = ascending aortic aneurysms associated with bicuspid aortic valve; MED = medial; POST = posterior; LAT = lateral; ANT = anterior.

			AsAA-QAV	AsAA-TAV	AsAA-BAV
**MED**	Failure stiffness (MPa)	LON	0.579	0.630	0.661
		CIR	0.676	0.891	0.717
	Thickness (mm)		2.41	2.256	1.714
**ANT**	Failure stiffness (MPa)	LON	0.698	0.775	0.667
		CIR	0.889	0.941	0.932
	Thickness (mm)		2.06	1.64	1.498
**LAT**	Failure stiffness (MPa)	LON	1.034	1.145	1.087
		CIR	0.911	1.250	0.905
	Thickness (mm)		2.212	1.78	1.306
**POST**	Failure stiffness (MPa)	LON	0.913	0.667	1.083
		CIR	1.509	0.814	1.384
	Thickness (mm)		2.064	1.574	1.586

**Table 3 jcm-11-04897-t003:** The content of collagen and elastin in the AsAA-QAV, AsAA-TAV, and AsAA-BAV. AsAA-QAV = ascending aortic aneurysms associated with quadricuspid aortic valve; AsAA-TAV = ascending aortic aneurysms associated with tricuspid aortic valve; AsAA-BAV = ascending aortic aneurysms associated with bicuspid aortic valve.

		AsAA-QAV	AsAA-TAV	AsAA-BAV
**Collagen (%)**	Intima	15.925	23.227	18.119
	Media	17.29	20.689	20.878
	Adventitia	34.612	43.273	43.985
	Total	19.95	26.28	23.09
**Elastin (%)**	Intima	24.337	19.297	34.682
	Media	31.681	28.83	31.142
	Adventitia	7.175	4.96	9.995
	Total	27.37	21.59	28.37

## Data Availability

The data presented in this study are available on request from the corresponding author. The data are not publicly available due to restrictions ethical.

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
