# Peer review of "What Are the Biomechanical Properties of an Aortic Aneurysm Associated with Quadricuspid Aortic Valve?"

_jcm, 2022, doi:10.3390/jcm11164897_

Round 1

Reviewer 1 Report

GENERAL COMMENT

Congratulations to the author for the organization of the in-vitro analysis of the samples, which is always an added value for the analysis of clinical findings and for the understanding of the physiopathology of diseases. In this context, being a quadricuspid valve a very rare anomaly of the aortic valve, the initial question on the biomechanical properties of an aneurysm associated with a quadricuspid valve could be interesting, but being based on the analysis of a single patient certainly cannot give definitive conclusions and scientific evidence.

DETAILED COMMENTS AND SUGGESTIONS

- Line 27-28:  In the introduction you can report the diameter ranges for the treatment of ascending aortic aneurysms.

- Line 94-96: in the previous part of the discussion (line 91 to 94) referring to table 1, you talked about the CIR e LON direction in the LAT quadrant of AsAA-TAV sample, at the line 95 when you refer to both AsAA-BAV and AsAA-QAV samples, do you always mean LAT quadrant? please specify the part of samples. Furthermore at line 95-96 you affirm that "the LON direction can observe a significantly greater failure stress than the LON direction", I think that the correct sentence is "...than the CIR direction", if you refer to LAT quadrant. Please clirify this concept.

COMMENTS TO AUTHOR

It would be useful and interesting to quantify the different elastin and collagen content by comparing the three different layers of aortic wall (intima, media and adventitia). 

Author Response

Thank you for your comments. Please kindly find our reply attached. 

Reviewer 2 Report

In this study the authors evaluate biomechanical properties in a quandricuspid aortic valve and compare therim with an bicuspid and tricuspid aortic valves. They found a quite similar biomechanical properties between aortic aneurysms with quadricuspid aortic valve and with bicuspid aortic valve. The study in interesting however some comments have to be adressed-A better decription of the 3 patients could be useful; age, risck factors as hypertension, medications, the funcioning of tha valve, (stenosi? Regurgitation?, )

-How did the authors evaluted the flow direction by CMR (4d flow?) could the authoes provide images of the flow directon for the 3 valves?

-  How did the authors manage to evaluate the strain? Please provide a more
accurate description

- The authors reported as a result the stress as a function of stiffness, you
should report the classic stress-strain curves as well

- It is unclear if the histology was performed before or after the
tensile tests

Author Response

(The authors gave the same response as above.)

Reviewer 3 Report

The research experiment arouses curiosity, however, it concerns a very small group. The methodology of the experiment is interesting. It is known that a bicuspid aortic valve (BAV) is a risk factor for aortic dilation and dissection. Therefore, the guidelines take into account a smaller size of the ascending aorta and aortic arch replacement in patients with BAV compared to patients with tricuspid aortic valve. The likelihood of treating a patient with a quadricuspid aortic valve is very, very small.

Author Response

(The authors gave the same response as above.)
